# Knowledge Source Rankings for Semi-Supervised Topic Modeling

**Justin Wood \*, Corey Arnold and Wei Wang**

Deparment of Computer Science, University of California, Los Angeles, CA 90095, USA;
cwarnold@ucla.edu (C.A.); weiwang@ucla.edu (W.W.)
\* Correspondence: juwood03@ucla.edu; Tel.: +1-310-825-0060

**Abstract:** Recent work suggests knowledge sources can be added into the topic modeling process to label topics and improve topic discovery. The knowledge sources typically consist of a collection of human-constructed articles, each describing a topic (article-topic) for an entire domain. However, these semisupervised topic models assume a corpus to contain topics on only a subset of a domain. Therefore, during inference, the model must consider which article-topics were theoretically used to generate the corpus. Since the knowledge sources tend to be quite large, the many article-topics considered slow down the inference process. The increase in execution time is significant, with knowledge source input greater than $10^3$ becoming unfeasible for use in topic modeling. To increase the applicability of semisupervised topic models, approaches are needed to speed up the overall execution time. This paper presents a way of ranking knowledge source topics to satisfy the above goal. Our approach utilizes a knowledge source ranking, based on the PageRank algorithm, to determine the importance of an article-topic. By applying our ranking technique we can eliminate low scoring article-topics before inference, speeding up the overall process. Remarkably, this ranking technique can also improve perplexity and interpretability. Results show our approach to outperform baseline methods and significantly aid semisupervised topic models. In our evaluation, knowledge source rankings yield a 44% increase in topic retrieval f-score, a 42.6% increase in inter-inference topic elimination, a 64% increase in perplexity, a 30% increase in token assignment accuracy, a 20% increase in topic composition interpretability, and a 5% increase in document assignment interpretability over baseline methods.

**Keywords:** topic modeling; PageRank; semisupervised learning

## 1. Introduction

The world is overflowing with text. This ever-growing resource has the ability to capture thoughts, ideas, and understanding. To extract, connect, and summarize relevant electronic text records has the potential for knowledge discovery and new understandings. One example is the electronic health record which often contains important raw text information regarding a patient as documented by a physician. The electronic health record is increasing rapidly as technology is integrating itself into the patient physician interaction. To be able to deliver information quickly and accurately to a physician can help ease the burden and lesson the mistakes that a primary care physician can make when dealing with the increasing pressure from seeing too many patients in too little time.

The enormous amount of textual data makes it impossible for people to manually undertake the task of processing all the information. Computational techniques must be developed that can overcome the challenges faced by working with a very large set of free-text input. This work presents approaches that seek knowledge discovery from a large input of text documents. We focus on the task of summarizing corpora to provide a set of topics describing the general themes.

The task of extracting topics comprises the field topic modeling. In this domain different approaches exist with similar aims. Matrix-based approaches such as latent semantic

analysis (LSA) may be used to reduce dimensionality of corpus and highlight more important words [1]. A more Bayesian technique involves assuming a generative model over the corpus and then discovering the component of the generative model through Bayesian inference [2]. More recently, the focus shifted towards deep learning-based approaches which seek to use neural networks for topic discovery. Undoubtedly combinations of these subdomains will yield improved results over running any model in isolation [3]. For the desiderata of using topic models to convey information to an application, it is paramount that the topics be interpretable and helpful if the topics are labeled.

Although lately topic modeling research seems to be directed towards neural topic modeling (NTM) [4–6], traditional Bayesian-based topic models (BTM) offer a viable alternative to deep learning approaches. Bayes approaches may be preferable when (1) using commodity or legacy hardware, as the NTM often requires a more complex setup (such as utilizing a graphics processing unit [GPU]), (2) a document-to-topic ($\theta$) distribution is needed, since for the NTM, $\theta$ is often associated with a batch parameter and reused for multiple documents [4–6], and (3) for more interpretable topics [7,8] since the high perplexity of the NTM may lead to lower interpretability [8] and the recent work challenges the goodness of traditional pointwise mutual information (PMI) based interpretability scoring often reported in NTM results [7,9]. The latter scoring method [9] may be the preferred approach to take for estimating interpretability of topic models, however we take direct human-based scoring to be a stronger approach to evaluate interpretability.

The traditional probabilistic topic model outputs a distribution of numeric topics for each document and a distribution of words for each numeric topic [2]. These latter distributions comprise the "topics" in topic modeling. As such, a "topic" is just a distribution over words with a numeric label. However, the numeric label fails to summarize the distribution semantically. Semantically labeling each topic gives the end user a quick understanding of what each topic represents, improving the interpretability [10]. These labels can also be used in downstream processes such as graph-based summarization systems [11,12], consensus building [13] and scene identification [14]. However, assigning an accurate label to a topic is no trivial task.

To assign semantic labels to topics, one can run an unsupervised topic model and then choose labels after inference [15–21]. However, this can lead to problems with the topics themselves as the clusters tend to combine two or more semantically different topics [10]. For example (Adapted from [10] with permission), suppose a news corpus that consists of two articles is given by documents $\mathbf{d_1}$ and $\mathbf{d_1}$ each with three words:

$$\mathbf{d_1}\text{—pencil, pencil, umpire}$$
$$\mathbf{d_2}\text{—ruler, ruler, baseball}$$

Latent Dirichlet allocation (LDA) [2], with the traditionally used collapsed Gibbs sampler, standard hyperparameters and the number of topics ($K$) set as two, would output different results for different runs due to the inherent stochastic nature. It is possible to obtain the following result of topic assignments:

$$\mathbf{d_1}\text{—pencil}^1, \text{pencil}^1, \text{umpire}^2$$
$$\mathbf{d_2}\text{—ruler}^2, \text{ruler}^2, \text{baseball}^1$$

But these assignments to topics differs from the ideal solution that involves knowing the context of the topics in which these words come from. If the topic modeling was to incorporate prior knowledge about the topics "School Supplies" and "Baseball", then a topic modeling process will more likely generate the ideal topic assignments of:

$$\mathbf{d_1}\text{—pencil}^2, \text{pencil}^2, \text{umpire}^1$$
$$\mathbf{d_2}\text{—ruler}^2, \text{ruler}^2, \text{baseball}^1$$

and assign a label of "School Supplies" to topic 1 and "Baseball" to topic 2.

A second approach to semantic topic labeling involves using a supervised input set and showed the ability to label the topic as necessary [22–25]. This approach requires many

labeled input that may be time-consuming or expensive to acquire. To allow for a labeled input set that is easier to obtain, semisupervised topic models [10,26–29] use existing knowledge sources as semisupervised input to label topics. The knowledge sources consist of articles turned into distributions and can be transformed into knowledge source topics ($\hat{\phi}$). Any generative model which utilizes $\hat{\phi}$ is dependent on a subset of labeled data, and thus we refer to this type of topic modeling as semisupervised topic modeling. To further illustrate the concepts of the knowledge source and semisupervised topic modeling, consider the following simple example. At the time of this writing, if we open a web browser and go to Wikipedia (https://en.wikipedia.org/w/index.php?title=Grape&oldid=908871054, accessed on 1 December 2021) and search for "grape", the returned article ($\hat{A}$) would start with the following text:

*A grape is a fruit, botanically...*

If we take the above to be the full article, then the knowledge source topic ($\hat{X}$) for "grape" can be formed by taking a count of each word ($\hat{w}$) in the article and dividing each word by the total number of words. For the "grape" example, the knowledge source topic is the probability vector $[\frac{2}{6}, \frac{1}{6}, \frac{1}{6}, \frac{1}{6}, \frac{1}{6}]$ with the index of the probability vector mapped to the word vector [a, grape, is, fruit, botanically].

If we continue the above for a set of articles from Wikipedia, the set of articles becomes the knowledge source (*KS*). We follow the above procedure from theknowledge source to get a set of knowledge source topics. These knowledge source topics are then used in the corpus's theoretical generative model. During inference, the topic model takes as input a set of knowledge source topics that may or may not be used in the final output of topics. Because the output is dependent on a subset of labeled data, we refer to this type of topic modeling as semisupervised topic modeling. One drawback of semisupervised topic modeling is the excess knowledge source topics used as input. Since there is a more relaxed constraint of not needing to know precisely which knowledge source topics are relevant to a corpus, there tend to be many knowledge source topics ultimately discarded. Existing approaches used to determine which topic to discard are based on counting or some form of clustering. However, counting is problematic because it is too simple and often discards importantknowledge source topics due to not having a high count. In this context, we take important topics to be topics which are used in the generative model of the corpus. Even worse is clustering, which only considers some distance metric between the topics and does not consider how many assignments of words were made to the topic. We illustrate these concepts using a simple case study.

*1.1. Case Study*

We are given the task of labeling patient notes from a small set of electronic health records. Given that we know we are in the medical domain, we suppose all possible and relevant topics for any patient note to be in the following set:

$\hat{\mathbf{A}}_1$—cancer, cancer, tumor, chemotherapy
$\hat{\mathbf{A}}_2$—heart attack, heart, attack chest
$\hat{\mathbf{A}}_3$—dementia, brain, memory, dementia
$\hat{\mathbf{A}}_4$—diabetes, blood, sugar, insulin

Next, we wish to obtain topics and corresponding labels for a corpus of two documents $\mathbf{d}_1$ and $\mathbf{d}_1$, given as:

$\mathbf{d}_1$—cancer, chest, attack
$\mathbf{d}_2$—tumor, heart, chemotherapy

A good semisupervised topic model would start by considering the entire knowledge source of ($\hat{\mathbf{A}}_1, \hat{\mathbf{A}}_2, \hat{\mathbf{A}}_3, \hat{\mathbf{A}}_4$) but would eventually end up with document-token to topic assignments of:

$\mathbf{d_1}$—cancer[1], chest[2], attack[2]
$\mathbf{d_2}$—tumor[1], heart[2], chemotherapy[1]

With topic 1 (after the topic model interference is complete) mapped to $\hat{\mathbf{A}}_1$ and topic 2 mapped to $\hat{\mathbf{A}}_2$. Since $\hat{\mathbf{A}}_1$ and $\hat{\mathbf{A}}_2$ are referenced in the final document-token assignment, we consider these *relevant* or *important* topics. Additionally, since $\hat{\mathbf{A}}_3$ and $\hat{\mathbf{A}}_4$ were not referenced by any document-token assignment to topic, we delegate these to be *discarded* topics.

It is essential for the semisupervised topic model to determine which topics are relevant and which topics to discard. What is needed is some way to rank the topics by order of importance to a corpus. A better ranking of topics can select the relevant topics and discard the less important ones. Counting can be used for ranking, but this leads to the problems discussed previously. One method for ranking which has already shown promising results is PageRank [30]. PageRank finds the importance of a node by considering the importance of the connecting neighbors in a recursive fashion. This approach helps determine the importance of websites in the world wide web.

With the success of PageRank in the world wide web, it is a natural approach to apply the techniques of PageRank to other ranking problems, such as the ranking of article-topics. The main obstacle of using PageRank for knowledge source rankings is representing the knowledge source as a graph consisting of nodes and edges. In most cases, a knowledge source consists of a collection of articles, i.e., Wikipedia articles corresponding to MedlinePlus (https://www.nlm.nih.gov/medlineplus/ (accessed on 1 December 2021)) headings. However, there are knowledge sources that already take the form of a graph, such as the Unified Medical Language System (UMLS) (https://www.nlm.nih.gov/research/umls/ (accessed on 1 December 2021)). Ontologies and other compendia exist that take the form of entities as nodes and relationships among entities as edges. For these cases we still need to determine a way to effectively rank the nodes and edges which is applicable in the context of semisupervised topic modeling.

Still, with the desiderata to increase applicability, we must consider how to rank existing article-based knowledge sources. This paper presents a novel way to aid topic models that already have a knowledge source associated with the corpus. Our technique applies to both graph-based and article-based knowledge sources. When we have both a graph and article-based knowledge source, we can take the topic labels from the article headings and emphasize these nodes in the graph-based knowledge source. When comparing the results after ranking, we can select the subset of nodes corresponding to article labels. We also formulate similar approaches for article-only and graph-only knowledge sources.

As we show in the results section, knowledge source rankings represent a significant improvement over counting for determining which topics to discard. However, even with a perfect partitioning of important and discarded topics, we are still limited in the amount of curated knowledge we can add into the semisupervised topic model [10]. At a knowledge source size of just 1000 article-topics, the inference iteration times become too high to be practical [10]. To further improve the applicability of our model, we aim to allow any input knowledge source regardless of size. Our solution is to rank the article-topics using our ranking method preinference and filter out low scoring article-topics. We can then input the filtered knowledge source into the semisupervised topic model and proceed as usual.

Knowledge source rankings are not only limited to preprocess filtering. The rankings are also applicable during topic modeling inference to help existing semisupervised algorithms determine which topics should be removed. We can also use knowledge source rankings in a stand-alone topic model or in the generative model alongside existing semisupervised topic models.

The intuition behind our ranking approach is like that of TextRank [31]. This established method ranks sentences in a document to determine a sentence used to summarize the document. Similarly, and with some modifications, we should be able to develop a technique to determine a ranking of article-topics relevant to a corpus. Additionally, knowledge source preprocess filtering has already been shown to improve text-related tasks [32]; and knowledge sources rankings utilize a graph representation to incorporate

outside information. Similar outside is already established to be helpful in text classification tasks [33,34], while graph representations can yield improved results as well [35].

### 1.2. Research Objectives

Semisupervised topic models showed the ability to improve upon traditional topic modeling in two ways: (1) an increase in interpretability and while not significantly impacting perplexity [10] and (2) the labeling of topics [10,26,27]. The first improvement is particularly significant since it was previously thought that improving interpretability ultimately leads to decreased perplexity [8]. So then why hasn't semisupervised topic models been widely adopted as a standard for every topic modeling? By design, the semisupervised input is effortless to obtain for most corpora. With no significant impact on perplexity, why not have more interpretable topics that are labeled? It seems that in most applications of topic modeling, this could only help. One reason may be due to the high execution time. At large semisupervised input sizes, the models become unfeasible. It is our objective to resolve this inadequacy. By removing the burden of high execution times while still maintaining the benefit of semisupervised models, we hope this topic modeling technique takes a step toward being the approach used in all topic modeling. Additionally, we seek to use the same techniques to speed up the execution of semisupervised topic models to further improve the interpretability and perplexity of these models. At the end the research paper, we hope to give existing consumers of semisupervised topic models another tool that can improve execution time, perplexity and interpretability; and to compel any topic modeler that semisupervised topic models are an effective enhancement to existing topic models on just about every dataset.

### 1.3. Article Outline

The rest of this article is organized as follows: in Section 2 we give a few motivating examples to help understand the intuition behind our ranking approach. In Section 3 a more extensive overview of related works and background into our problem domain is presented. Section 4 provides the details of our approaches named *KnowledgeRank* and Rank-LDA.

The results of our approaches compared against various baseline methods and datasets are given in Section 5. We provide a brief discussion of our approach, their results and implications in Section 6. And finally we conclude the article in Section 7.

## 2. Motivating Examples

We provide a few small examples to help understand the intuition behind using ranking algorithms for semisupervised topic models.

### 2.1. Graph-Based Knowledge Sources

The proposed ranking algorithm allows for the inclusion of graph-based knowledge sources into the semisupervised topic modeling process. Current methods only allow for article-based knowledge sources [10,26,27]. For example, suppose we are working with a corpus of PubMed (https://www.ncbi.nlm.nih.gov/pubmed/ (accessed on 1 December 2021)). articles, and we observe the word acetylsalicylic acid (commonly known as aspirin). We are now trying to classify this word as belonging to either *Cerebral infarction* or *Alzheimer's disease* using an article-based knowledge source derived from Wikipedia. However, neither the article for *Alzheimer's disease* https://en.wikipedia.org/w/index.php?title=Alzheimer%27s_disease (accessed on 1 December 2021) nor *Cerebral infarction* https://en.wikipedia.org/w/index.php?title=Cerebral_infarction (accessed on 1 December 2021) contains the word *acetylsalicylic acid* (as well as aspirin), leaving the model to choose the topic assignment from outside the knowledge source. However, suppose we were to leverage the graph-based knowledge source UMLS. In that case, we have a direct connection (C0007785 RO/may_be_prevented_by C0004057) between *Cerebral infarction* and *acetylsalicylic acid*—whereas none exists between *acetylsalicylic acid* and *Alzheimer's*

*disease*. This extra information can help to classify *acetylsalicylic acid* to *Cerebral infarction* over *Alzheimer's disease* at a more accurate percentage (Based on a PubMed MeSH term search of *acetylsalicylic acid*) and *cerebral infarction* versus *acetylsalicylic acid* and *Alzheimers disease*, yielding 488 to 58 results respectively.

### 2.1.1. Overlapping Topics

Another advantage of semisupervised topic ranking comes from leveraging information from overlapping topics. In this example, suppose we try to classify $w_1$ as belonging to $t_1$ or $t_2$, and $w_1$ is not in either $t_1$ or $t_2$'s knowledge source article. However, a third topic, $t_3$, contains $w_1$ and $w_2$, which $t_1$ shares. Furthermore, $t_2$ does not share any words with $t_3$. Thus, ranking can help prefer $t_1$ over $t_2$ as the score is propagated based on distance. However, other methods: counting, Gibbs-based, etc., cannot give such an advantage.

### 2.1.2. Discarding Topics

At some point, the topic model must choose to discard topics assumed not to be used in the generative model. Existing methods use counting, assuming that if a topic was used in the generative model, then there will be more word assignments to that topic than a topic not used in the generative model. However, this may not be the best way to eliminate topics. Consider the example shown in Figure 1. Here, we modeled assignments of words to topics as a graph with a word having an outgoing edge to a topic if that word is assigned to that topic. If we must discard one topic out of the existing topic set, counting would choose $t_4$. However, a better topic to discard would be $t_1$ since the words assigned to $t_1$ are shared words that could easily be from other topics. Ranking would consider the context of the words assigned to $t_4$ to choose $t_4$ over $t_1$.

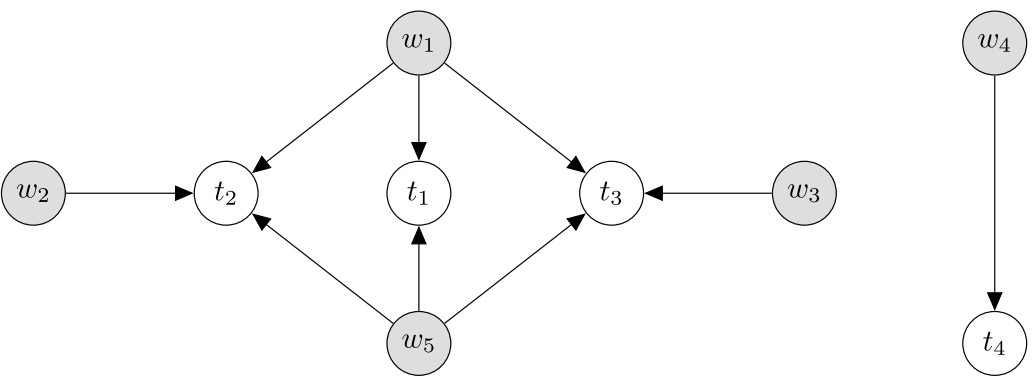

**Figure 1.** An example graph representation of word and topic assignments.

## 3. Background

### 3.1. Semisupervised Topic Modeling

A known weakness of traditional topic modeling (LDA) [2] is a lack of interpretability [8]. Given a set of topics, a human annotator can often have difficulty identifying a label for the topics. One solution to increase interpretability comes in the form of semisupervised topic modeling.

Some forms of semisupervised topic modeling combine unsupervised models with supervised models to classify documents [36,37]. Although an adoption could apply to modeling topics [38], the supervised portion still requires an input set that may be expensive or challenging to obtain. Other forms of semisupervised topic modeling take the semisupervised input directly from the input itself [39]. While the approach is effective, such as in the domain of sentiment classification, this may not apply to our domain since they do not incorporate outside knowledge and may lead to problems when the input does not contain enough information to form meaningful topics [10].

We restrict our study to semisupervised topic modeling which contains some easy to obtain superset of labeled topics given as input to latent Dirichlet allocation (LDA) [2]. This labeled topics is often referred to as a knowledge source.

We formalize the *knowledge source* (*KS*) as:

$$\hat{\phi}_j \sim f_\phi(\hat{X}, KS, \hat{A}_j, \beta) \tag{1}$$

$$KS = (\hat{A}_1, \hat{A}_2, \dots, \hat{A}_{\hat{B}}) \tag{2}$$

$$\hat{X}_j = f_{\hat{X}}(\hat{A}_j) \tag{3}$$

$$\hat{A}_j = (\hat{L}_j, \hat{w}_{1,j}, \hat{w}_{2,j}, \dots, \hat{w}_{\hat{G}_j,j}) \tag{4}$$

With $\hat{G}_j$ being the word count of article-topic $\hat{A}_j$, $\hat{L}_j$ the article label, $\hat{w}_{i,j}$ as the $i$th word in $\hat{A}_j$, $\beta$ is a Dirichlet distribution hyperparameter and $f$ being functions that the model determines.

This labeled topic input is assumed to be part of the generative model. Before generating the corpus, we determine the total number of topics ($K$) and vocabulary size ($V$). For each *topic*, we sample from a Dirichlet distribution that may or may not be influenced by an individual knowledge source topic. If a knowledge source topic influences the topic, the *topic label* becomes the article's title from which the knowledge source topic was created ($\hat{L}$). Each *document* in the corpus is generated by first sampling a *topic* from a discrete distribution of size $K$. After the topic is sampled, a *word* is chosen by sampling from the topic's discrete distribution ($\phi$) of size $V$.

During the corpus generation, some topics are formed using the technique from LDA, while a set of others are drawn from a function of the labeled input data. This function can place a Dirichlet prior over the vocabulary [27,40]; however, this tends to lose any semblance from the labeled input data. Since the labeled input data tend to be highly interpretable, a more interpretable approach involves assuming the labeled input data are topics themselves [26,41]. The labeled input often comes in the form of documents describing a topic, referred to as a knowledge source. These documents are formed into histograms and directly turned into distributions representing the histograms. This approach increases interpretability but can be too rigid in representing a labeled topic [10].

A third approach, Source-LDA, involves a compromise between the previous two approaches [10]. Source-LDA draws its labeled set of distributions directly from the knowledge source; however, it assumes these histogram counts to be the hyperparameters to a Dirichlet distribution. By allowing the histogram counts as parameters into the Dirichlet distribution, variance is allowed on the labeled topic set and is more adaptive to the data. This balance between interpretability and adaptability allows for Source-LDA to outperform existing methods of semisupervised topic modeling.

The plate diagram for Source-LDA is highlighted in Figure 2, and a similar adaption can be imagined for any semisupervised input model [26,27]. The variables in Figure 2 are explained in Table 1. Since the labeled topics are part of the generative model, the inference must consider these new variables for any semisupervised topic model. A general Gibbs sampler [42] can be built using the sampling condition given as:

$$P(z_i{=}j|\vec{z}_{\text{-}i}, w) \propto d^L(n^w, n^d, \beta, \alpha, V, i, w, D) \tag{5}$$

with $d^L$ representing a posterior sampling density [42,43] and for all $i > T$:

$$P(z_i{=}j|\vec{z}_{\text{-}i}, w) \propto d^S(n^w, n^d, \beta, \alpha, V, i, w, D, KS) \tag{6}$$

where $\vec{z}$ is a vector of topic assignments, $i$ the current token, $w$ is a matrix representation of all the words, $n^w$ is a matrix of topic word counts. $\beta$ is the symmetric hyperparameter for the word to topic mixtures, $V$ is the size of the vocabulary, $d^S$ is a posterior sampling density specific to the model, $n^d$ represents the matrix of document topic counts, $\alpha$ is the symmetric

hyperparameter for the topic to document mixture, *K* is the count of all nonlabeled topics, and *T* is the total number of topics.

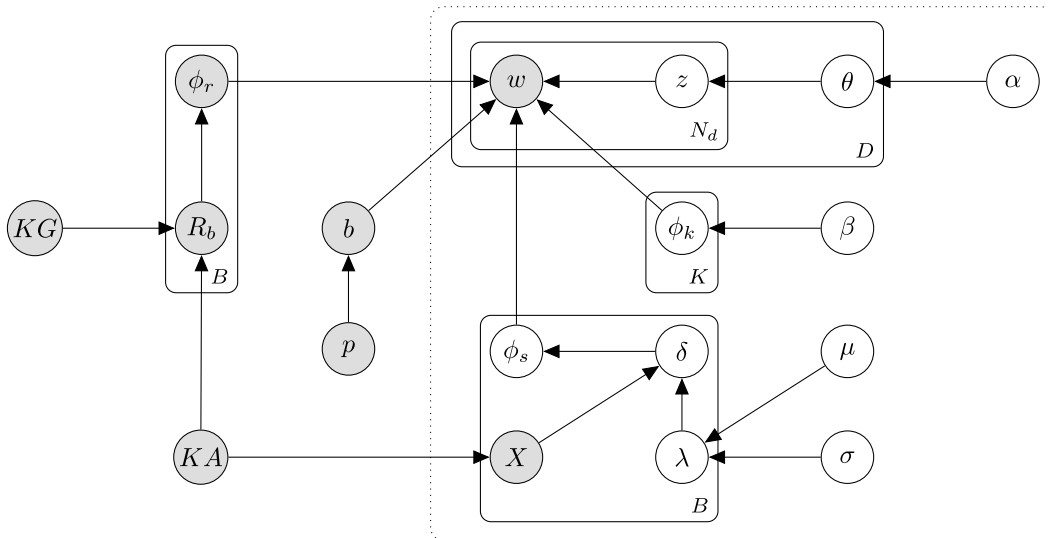

**Figure 2.** Plate notation for Rank-LDA. Dashed box represents Source-LDA.

**Table 1.** Notations used in Source-LDA.

| Symbol | Description |
| --- | --- |
| $w$ | A word in a document of size $N_d$ |
| $z$ | The topic corresponding to $w$ |
| $\theta$ | A distribution over topics for each $d_i \in D$ documents, parameterized by $\alpha$ |
| $\alpha$ | The Dirichlet hyperparameters for each $K$ topics |
| $N_d$ | The number of words in $d_i \in D$ documents |
| $D$ | The number of documents in the corpus |
| $\phi_k$ | A distribution over words for each $k \in K$ topics, parameterized by $\beta$ |
| $\beta$ | The Dirichlet hyperparameters for each $w$ words |
| $B$ | The number of knowledge source topics |
| $K$ | The number of latent topics |
| $\phi_s$ | A distribution over words for each $b \in B$ topics, parameterized by $\delta$ |
| $\delta$ | The Dirichlet hyperparameters for each word in $b \in B$ topics. The value is a result of a function applied to $X$ and $\lambda$ |
| $\mu$ | The mean to the normal distribution |
| $KA$ | An article-based knowledge source |
| $X$ | The count of each word in $a \in KA$ knowledge source article |
| $\lambda$ | A latent number that signifies how far $\phi_s$ deviates from the corresponding frequency distribution |
| $\sigma$ | The standard deviation to the normal distribution |

### 3.2. Pagerank

PageRank [30] uses an iterative algorithm to determine a ranking for a node. Additional variables are added to the single-step calculation due to some nodes' potential to have no outbound edges or no inbound edges. These additional variables lead to the calculation of a single step ranking (*R*) for a given node *m* as:

$$R(m) = \frac{1-d}{|N|} + d \times \sum_{n \in I(m)} \frac{R(n)}{|O(n)|} \tag{7}$$

$I(x)$ and $O(x)$ return a set of all the inbound and outbound nodes connected to node *x*, respectively, and *N* is the set of all nodes. The variable *d* represents the damping factor.

An interpretation of PageRank can be that of a random surfer. The random surfer starts at any one web page randomly in the world wide web and either clicks a random link with a probability of *d* or goes to any other page randomly in the world wide web. The result of the ranking is the probability that the random surfer will visit a page.

### 3.3. Ranking Algorithms

The original PageRank algorithm was used to determine the importance of a given web page [30]. This ranking was shown to be an excellent way to rank the importance of a word in a sentence [31] and determine trustworthy websites [44]; however, they are only applicable to their respective domains. The integration of topics into PageRank can help weight the edges used in PageRank. This weighting is accomplished using topic vectors determined by topic lists [45]. This method, while effective in improving the results of PageRank, yields no improvement in topic modeling. PageRank was also used to add weighting to some classification and topic tasks [46–48]. As in TextRank, vanilla PageRank works to find an importance ranking among connected components. These and other approaches help form a motivation for our work, but they do not offer insights to how they may rank knowledge sources.

### 3.4. Topic Labeling

Topic labeling can be done in the postprocessing stage [15,16] by comparing the topic distributions with some predetermined knowledge source. The drawback of these approaches is that the topics tend to cluster nonsemantically related words [10]. Supervised approaches allow predetermined labels to be assigned to clustered topics [23–25]. These approaches can assign an entire document a label [23,24] or assign multiple labels to a document [25]. Supervised techniques are often dependent on an extensive collection of labeled data that may be expensive or time-consuming to obtain.

A balance between after-inference topic labeling and supervised topic labeling comes from semisupervised topic modeling, previously discussed in Section 3.1. These methods use some form of labeled input, much like supervised topic modeling; however, the input is much easier and cheaper to obtain.

Different approaches outside of Bayesian modeling can be performed to label topics or similar tasks [49,50]. One example is mapping the vectorized tokens against the tokens of the corpus [49]. After the corpus tokens are mapped to vectors, classification is run using a deep learning model such as a neural network. This approach yields good results when the input is labeled and enough training data exist to build a supervised model. When there is not enough labeled data, the supervised model may yield poor results.

## 4. Methods

With the desiderata to leverage graph-based knowledge sources in topic modeling, we must first model the semisupervised input in a way that maximizes the effectiveness of the ranking. We introduce our approach, KnowledgeRank, for constructing a graph-based representation of a knowledge source for ranking the appropriate nodes and edges. Variables used in KnowledgeRank are summarized in Table 2.

**Table 2.** Explanations for variables used to describe the methodology behind KnowledgeRank.

| Symbol | Description |
| --- | --- |
| $R_g(n)$ | The rank score for a node $n$ in graph $g$ |
| $I(m)$ | The set of all nodes with incoming edges into node $m$ |
| $O(n)$ | The set of all nodes with incoming edges originating from node $n$ |
| $C_n$ | The count of word $n$ in a corpus |
| $\hat{D}$ | A corpus |
| $d$ | Damping factor |
| $\zeta_g$ | An input parameter over the interval $[0, 1]$ specifying the ranking importance of frequent words in a corpus with corresponding knowledge source graph $g$ |
| $N$ | The set of all nodes |
| $P(w_i|t)$ | The probability of word $w_i$ given topic $t$ |
| $S(t, w_i)$ | The set of nodes in the shortest path from node $t$ to node $w_i$ |
| $X(m, n)$ | The number of times word $n$ appears in topic $m$'s knowledge source article |
| $z_i$ | The $i$th topic assignment |
| $\vec{z}_{-i}$ | A vector of all topic assignments minus the $i$th assignment |
| $b$ | A variable representing the draw from a Bernoulli distribution |
| $n_{-i,j}^{w_i}$ | The number of assignments of word $i$ to topic $j$ minus the current assignment |
| $n_{-i,j}^{(\cdot)}$ | The number of assignments to topic $j$ minus the current assignment |

*4.1. Graph-Based Knowledge Sources*

In cases where the only semisupervised input set is already in the form of a graph, we can simply use the given structure as the model for KnowledgeRank. However, what is not entirely clear is how to obtain the labels. Many ontologies or other compendia consist of concept nodes that can be used as labels for topics and noisy word nodes that would be inappropriate labels for a given topic. For example, in the neuroscience information framework (NIF) ontology, a given node may correspond to the word "of" which obviously would not be a good label for any topic. These less applicable words have to do with knowledge sources containing parts of speech or commonly used words in their respective texts. This curated data source can still be helpful for topic models, but we must first find the appropriate labels.

Graph-based ranking models have already established the ability to find the most important word in a sentence [31]. It follows that similar techniques can find the most important node from a set of nodes. If we apply the ranking algorithm to a knowledge source graph, we can determine the labeling for a topic based on the highest scoring nodes.

By applying the ranking in this way, we can obtain the most important nodes in the graph, but in some cases, we may want to let the corpus give us insight into the importance of a node. It is plausible that a word used more frequently in a corpus should be considered more important in the representative graph than one that is used very seldom. In other cases, this weighting is not so important. To account for these cases, we can augment the original PageRank formula to consider these weights and the associated importance of the weighting ($\zeta_g$) as:

$$w_1 = \sum_{n \in I(m)} \frac{R_g(n)}{|O(n)|} \tag{8}$$

$$w_2 = \sum_{n \in I(m)} \frac{C_n}{|D|} \cdot R_g(n) \tag{9}$$

$$R_g(m) = \frac{1-d}{|N|} + d \cdot [\zeta_g + (1 - \zeta_g) \times w_2] \cdot w_1 \tag{10}$$

$C_i$ is the count of word $i$ in the corpus $D$, and $\zeta_g$ is defined over the interval 0 to 1.

We can also use this information in the generative model itself. Given that we only have the graph-based knowledge source, we can construct a distribution over the vocabulary in a similar manner. We can form a distribution over the vocabulary by starting at a topic

label node, $t$, and normalizing the probability of arriving at each word in the vocabulary. The distribution can be calculated by considering the path a random surfer takes to each node with the restriction that the random surfer starts at each labeled node. This function is given as:

$$P_g(w_i|t) \propto \prod_{m \in S(t,w_i)} R_g(m) \tag{11}$$

The advantage of this approach is that the change required to infer the model's hidden variables can easily be adapted to any semisupervised topic model's Gibbs sampling equation. We can precompute the probabilities and then use the distributions the same way as a word distribution from an article-based knowledge source. In this approach, we add curated outside knowledge while still allowing LDA to cluster the topics.

### 4.2. Article-Based Knowledge Sources

For those knowledge sources consisting only of articles, we can model the articles into a graph and then run our ranking algorithm. Our approach connects each topic node to each corresponding source article word. Because frequent words in an article are assumed to be more important to topic identification, we would like to give these words more weight in our graph representation. We add this weighting by creating an edge (from topic to word) for each token in an article. For example, take the two histograms corresponding to a knowledge source article (article-topic) shown in Figure 3a. In this example, each $t_i$ represents a knowledge source topic label (or article heading) with each $w_i$ as a non-topic label word in knowledge source topic $i$. We model the edges as undirected resulting in $I(n) = O(n)$. Note that an article-topic can have in its article a word that is also a label for another article-topic. Also note that a word can be a non-topic label word (shows up in the body of the text) and the knowledge source topic label (the article heading) in the same knowledge source topic (such as $t_i$).

The change required to the ranking algorithm is the weighting of each node. This change gives us:

$$w_3 = \sum_{n \in I(m)} \frac{R_a(n)}{|O(n)|} \tag{12}$$

$$w_4 = \sum_{n \in I(m)} \frac{X(m,n)}{\sum_{m \in O(n)} X(m,n)} \cdot R_a(n) \tag{13}$$

$$R_a(m) = \frac{1-d}{|N|} + d \cdot [\zeta_a \times w_3 + (1 - \zeta_a) \times w_4] \tag{14}$$

where $\zeta_a$, a parameter defined between 0 and 1, lets us specify the importance of weighting the edges over a PageRank score, and $X(m,n)$ is the count of the number of token assignments word $n$ has in knowledge source topic $m$.

We can then use the graph-based representation in tasks mentioned in the graph-based knowledge sources section with this representation. This method would be beneficial in preprocessing to decrease some of the unimportant topics.

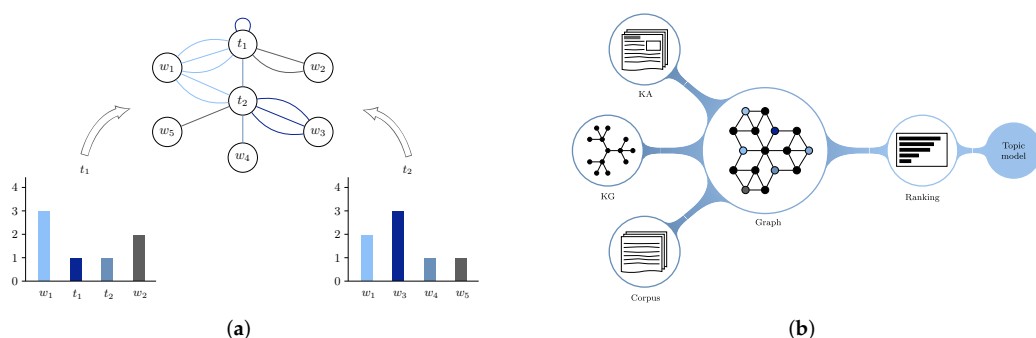

**Figure 3.** A graph-based representation of two topic ($t_1$, $t_2$) histograms corresponding to knowledge source articles (**a**) alongside a diagram representation of knowledge source rankings (**b**). In ranking diagram (**b**), *KA*, *KG*, and a corpus are used to construct a graph from left to right, used as input into a semisupervised topic model.

### 4.3. Graph and Article Knowledge Sources

Having both graph- and article-based knowledge sources brings a more extensive set of information into the topic model and thus can lead to better labeling of found topics. Given that we already have the graph form, we can apply the ranking algorithm to preprocess the existing knowledge source articles. We would want to let the corpus tell us about the importance of a word, but we also want to consider how important it is in the knowledge source article. For this, we make a change to the ranking calculation that allows for this weighting:

$$w_5 = \zeta_g \times w_1 + (1 - \zeta_g) \times w_2 \tag{15}$$

$$w_6 = \zeta_a \times w_3 + (1 - \zeta_a) \times w_4 \tag{16}$$

$$R(m) = \frac{1-d}{|N|} + d \cdot w_5 \cdot w_6 \tag{17}$$

We can use this ranking to perform all the tasks previously mentioned, such as preinference topic filtering, as we diagram in Figure 3b. Additionally, this ranking can be helpful in the inference stage of existing semisupervised topic models. During inference, the topic model must decide which topics to keep and which ones to discard. To determine which topic to discard, the algorithm considers a simple observable property such as the count of assignments to a topic. This decision can lead to problems such as when two related topics are used in a corpus, and thus one takes most of the overlapping word assignments. The topic with the smaller number of overlapping word assignments is then discarded. When using clustering algorithms, the same problem exists, limiting the similarity of two topics to a distance measure. Compared to clustering, using counts has more of an underlying intuition. We can use the ranking methods described previously as a third way of determining which topics to discard. After obtaining a ranking, we can simply remove an appropriate number of low-scoring topics.

Both knowledge sources can also be combined in a topic model that leverages the graph-based connections to increase the probability of words being assigned to the appropriate source topic when they do not appear in the knowledge source article. An incomplete assumption of article-based knowledge sources is that they contain every word for which the generative model would use to write about a particular topic, but this is certainly not the case. It is entirely possible that important words about a topic may not show up in a random document describing that topic. Graph-based knowledge sources can help add more information into the model. The generative process can be changed to allow for this synthesis of information. The change required to Equation (11) is:

$$P(w_i|t) \propto \prod_{m \in S(t,w_i)} R(m) \tag{18}$$

The graph-based knowledge source can also influence the word assignments by giving a graph-based distribution an influence that can be more or less than the article-based distribution depending on the data. To allow for this, we can place a Dirichlet prior over the selection between the models. Based on an input hyperparameter, the data will decide which distribution to select, and then we sample from this multinomial to determine which knowledge source is used to select the word. A more straightforward approach assumes that the vocabulary of the samples for the different types of knowledge sources is disjoint. This approach allows the generative model to sample the knowledge source choice variable (*b*) from the Bernoulli distribution, parameterized by *p*. During inference, *p* is easily observed and does not factor into the inference other than to determine which calculation to use.

As shown in Figure 2 (with the variables explained in Table 3), we can build a Gibbs sampler from the generative model. The choice variable, *b*, should be included in the Gibbs sampling and used to determine which distribution to sample from. The step sampling for $b = 0$ is the same as Equation (6). For $b = 1$, the step sampling is drawn from the proportional probability of [$P(z_i{=}j|\vec{z}_{-i})$ unchanged and omitted]. We take this approach to be Rank-LDA:

$$P(z_i{=}j|\vec{z}_{-i}, w_i, b{=}1) \propto \frac{n_{-i,j}^{w_i} + P(w_i|j)}{n_{-i,j}^{(\cdot)} + 1} \tag{19}$$

**Table 3.** Notations used in Rank-LDA.

| Symbol | Description |
| --- | --- |
| $\phi_r$ | A distribution over words for each $b \in B$ topics, parameterized by $R_b$ |
| $KG$ | A graph-based knowledge source |
| $R_b$ | The Dirichlet hyperparameters for each word in $b \in B$ topics influenced by KnowledgeRank |
| $p$ | Bernoulli distribution parameter |
| $b$ | Draw from the Bernoulli distribution parameterized by $p$ to determine which knowledge source $w$ is drawn from |

Rank-LDA is shown in Figure 2 as an extension to Source-LDA however a similar extension to any semisupervised topic model would result in a congruent construction. Rank-LDA uses the article-based knowledge source ($KA$) in two ways. The first being the original way used in the semisupervised topic model; the second is to provide supplemental support to the graph provided by $KG$. The intuition is that both $KA$ and $KG$ provide partial information about a topic and that combining them can only help. Additionally, by turning $KA$ into a graph, we take advantage of ranking over counting, which gives us the advantages discussed in the motivating examples. One disadvantage of this approach is that it does not consider the quality of the knowledge sources ($KA$ and $KG$), thus weighting them equally. A poor quality knowledge source could add noise, leading to less desirable results [51]. Knowledge source weighting and optimization are left as an open research area.

## 5. Results

Knowledge source rankings are applied in various experiments to show the utility of KnowledgeRank.

### 5.1. Datasets

To examine how well our algorithm performs across different datasets, we collected datasets across various domains and varying sizes. Details and metrics are provided in Tables 4 and 5. Links to download the datasets are provided in the data availability statement. The datasets can be partitioned into two sets: hierarchical and nonhierarchical. For the nonhierarchical datasets, we required a corpus with topics labeled by a human annota-

tor. Each dataset was taken from previous work on similar topic modeling tasks [52,53]. The datasets were preprocessed differently depending on the experiment; however all datasets were converted to lower case with nonascii characters removed. Additionally, to mitigate against stop words and obscure words we filter out the top and bottom 5% of occurring tokens. Outside of these standard data cleansing steps more details of data processing are provided in each experiment's experimental setup. The hierarchical datasets consist of parent, child relationship topic pairs. Each child was restricted to one parent, while each parent could have multiple children. Thus the network structure resembled a forest as opposed to a graph. More details about construction are given in the experimental setup for the hierarchal experiments (Section 5.6).

**Table 4.** Nonhierarchical datasets and their descriptions, article-based knowledge sources (KA), graph-based knowledge sources (KG), document count (D), and number of topics (K) used for evaluation of KnowledgeRank.

|  | Description | KA | KG | D | K |
|---|---|---|---|---|---|
| MeSH | Medical subject headings | Wikipedia | UMLS | 2000 | 56,326 |
| CiteULike-180 | Manually tagged scholarly papers | Wikipedia | WordNet | 182 | 1660 |
| FAO-30 | Manually annotated documents from the Food and Agriculture Organization of the UN. | Wikipedia | WordNet | 30 | 650 |
| SemEval-2010 | Scientific articles with manually assigned keyphrases | Wikipedia | WordNet | 244 | 3107 |
| Reuters-21578 | Manually labeled documents from the 1987 Reuters newswire | Wikipedia | WordNet | 21,578 | 2663 |

**Table 5.** Hierarchical datasets and their descriptions, article-based knowledge sources (KA), graph-based knowledge sources (KG), and number of topics (K), used for evaluation of KnowledgeRank.

|  | Description | KA | KG | K |
|---|---|---|---|---|
| MeSH | Medical subject headings | Wikipedia | UMLS | 130 |
| PhySH | Physics Subject Headings | Wikipedia | WordNet | 36 |
| ACM-2012 | ACM computing classification system | Wikipedia | WordNet | 4 |
| OAD-Wiki | Outline of academic disciplines | Wikipedia | WordNet | 70 |

*5.2. Execution Time*

For KnowledgeRank to be helpful in preprocessing, we seek to add a filtering approach that does not significantly add to the overall time needed to perform topic modeling. An execution cost that is minuscule compared to the time needed to complete Gibbs sampling of a corpus is ideal, given that execution times of semisupervised topic models can be quite expensive [10]. Any time increase in the topic modeling process will undoubtedly lead to a decrease in the usage of a semisupervised topic model.

We run KnowledgeRank as the preprocessing step on a dataset that consists of articles from Wikipedia corresponding to MeSH terms. We seek to obtain the best $K$ topics from a superset of $T$ knowledge source distributions. With $K$ taken as 100, 200, 500, and 1000 topics. $T$ also varies from 0 to 50,000 superset topics. Figure 4a shows that the execution time increases linearly with an increase of $T$. The different values of $K$ do not significantly impact the results, and even at extreme values of $K$ and $T$, the total execution time is relatively small; at 1.5 s, this is much less than the time taken to run state-of-the-art methods [10].

The same experiment was performed on each of the nonhierarchical datasets. To show the linearity of the execution times, we compare the average coefficient of determination of a linear function fit to the data against a quadratic function. The functions were fit using the least squares approach. As shown in Table 6, the results show more of a linear relationship than a quadratic relationship for the execution times of KnowledgeRank in preprocessing.

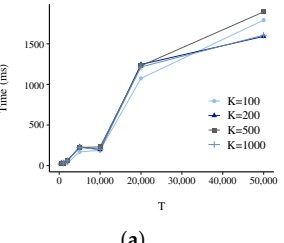

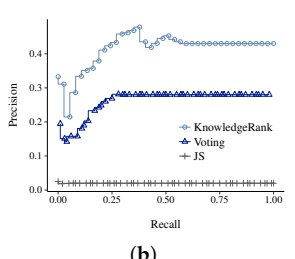

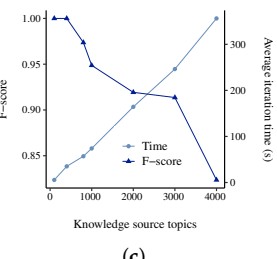

(**a**)                                                 (**b**)                                                 (**c**)

**Figure 4.** Results showing execution time for running KnowledgeRank (**a**), precision-recall curve for selecting topics used in generation of the corpus (**b**) and trade-off between execution time and F-score (**c**). All results shown are in preprocessing stage for MeSH dataset.

**Table 6.** Metrics describing execution of KnowledgeRank in preprocessing stage.

|  | Execution Time | | Preprocessing | | | Trade-Off |
| --- | --- | --- | --- | --- | --- | --- |
|  | $\bar{R}^2_{f(x)}$ | $\bar{R}^2_{f(x^2)}$ | $AUC_{rank}$ | $AUC_{vote}$ | $AUC_{js}$ | $r$ |
| MeSH | 0.905 | 0.768 | 0.418 | 0.264 | 0.02 | $-0.963$ |
| CiteULike-180 | 0.519 | 0.299 | 0.244 | 0.192 | 0.192 | $-0.529$ |
| FAO-30 | 0.344 | 0.186 | 0.269 | 0.238 | 0.238 | $-0.836$ |
| SemEval-2010 | 0.493 | 0.28 | 0.229 | 0.104 | 0.104 | $-0.821$ |
| Reuters-21578 | 0.437 | 0.236 | 0.223 | 0.164 | 0.164 | $-0.764$ |

### 5.3. Preprocessing

A proposed advantage of KnowledgeRank is the ability to appropriately determine which topics are used in the generation of a corpus. We show the utility of KnowledgeRank in this task by comparing it to baseline methods. We consider only baseline methods that require much less computation cost than that of topic modeling.

#### 5.3.1. Experimental Setup

We generate a corpus by first taking a random subset of MeSH article headings and combine them with each MedlinePlus article heading. For each article heading, we search Wikipedia for the corresponding article. If a query leads to no results or multiple results, we discard the article heading. The process results in 4300 found Wikipedia articles. Each Wikipedia article is then turned into a histogram over the set of words in the article. Given the histograms corresponding to Wikipedia articles, we generate a corpus of 2000 documents, each consisting of an average of 500 words using the Source-LDA generative model. The Source-LDA parameters are $K$, $\alpha$, $\mu$, and $\sigma$ set to 100, 0.5, 5, and 2, respectively. For KnowledgeRank, we take as input the SNOMED CT (http://www.snomed.org/, accessed on 1 December 2021) subset of the UMLS. The graph is filtered by removing any node whose corresponding string label does not occur in the corpus. We then run KnowledgeRank on the filtered graph. The first baseline method is based on voting, where one vote is cast to each topic for every word in both the corpus and the corresponding knowledge source article. A second baseline method is constructed by taking each document as a discrete distribution and scoring the likelihood of a topic existing in the corpus by comparing the Jensen–Shannon (JS) divergence. We then repeat this experiment for all datasets and record the area under the curve ($AUC$) of the precision-recall curve.

#### 5.3.2. Experimental Results

The corpus and knowledge sources are used to determine which topics are used to generate the corpus. Figure 4b shows the precision-recall curve for determining whether a topic was used in the corpus. KnowledgeRank outperforms the baseline methods significantly as the JS divergence baseline method has a hard time separating the mixtures, and voting is not refined enough to accurately capture the matching. Bringing into the

model the outside information of the UMLS allows for a more accurate determination of correct topics while doing so in a computationally efficient manner. Table 6 confirms that KnowledgeRank is consistently better in preprocess filtering than baseline methods.

### 5.4. Preprocessing Trade-Off

As shown in the previous experiment, KnowledgeRank can effectively filter out some topics from the knowledge source but cannot perform this task perfectly. Some filtered out topics could potentially be used to generate the corpus. A natural question to ask is whether the preprocessing is needed at all, since keeping all topics in the topic model allows the topic model to determine whether the topic is needed, based on an accurate Gibbs sampling [10].

The primary factor in deciding to use preprocess filtering is the amount of time it takes a semisupervised topic model to run entirely. This time is significant for existing semisupervised methods with a large corpus, approximation steps, and knowledge source size [10]. As shown in Figure 4c, for a corpus of 2000 documents averaging 500 words per document and $K$ set to 100 topics and 10 approximation steps, as the knowledge source increases, so does time. At the extreme end, one iteration takes over 350 s. It is simply not feasible to run the model on such an input size.

The solution is to reduce the knowledge source size using KnowledgeRank. But by doing this, we sacrifice some F-score. Figure 4c shows the trade-off expected when we filter out all but $K$ topics from the knowledge source before inference. As expected, as we increase the number of filtered topics, we inevitability decrease the F-score, as the ranking model has more choices to skew the filtering. This relationship is verified with the other datasets. The anticorrelation ($r$) is shown in Table 6 as the Pearson correlation coefficient.

### 5.5. Inference Pruning

Given that the input into the semisupervised topic model is a superset of topics, at some point, the topic model must decide which topics to keep and which topics to discard. Additionally, since $K$ unlabeled topics are thrown into the mix in the mixed models, a determination must also be made on these unlabeled topics.

KnowledgeRank can be used in these determinations by helping sort out which topics are best to keep around in a more in-depth manner than the current method of counting. The following experiment verifies this and compares its selections against clustering-based methods.

#### 5.5.1. Experimental Setup

A corpus was generated consisting of 2000 documents with an average of 500 words per document using 100 Wikipedia articles taken from MeSH subject headings. The Source-LDA generative algorithm was used to create the corpus from the 100 selected Wikipedia articles. The parameters for Source-LDA were $\alpha$, $\mu$, $\sigma$ set to 0.5, 0.7, and 0.3, respectively. We run Source-LDA with a knowledge source of 1000 medical subject headings with the generated corpus, inclusive of the 100 selected topics to generate the corpus. This process does not always yield incorrect decisions from Source-LDA using simple counting. Therefore, random permutations of the 1000-topic superset and 100 selected topic set were used as input into this process. The 100 and 1000 topic sets were sampled from a full MeSH and UMLS overlapping set of 8000 topics. Once a corpus and knowledge source were found, we log the decisions, count vectors, and $\phi$ distributions at each relevant step of the topic model and run the different methods to see if they can improve upon the decisions.

The decisions are made using KnowlegeRank and the established clustering algorithms: k-means clustering and density-based spatial clustering of applications with noise (DBSCAN) [54]. For KnowledgeRank, a graph was constructed using the counts as a weight from a word node and a topic node. If word $i$ was assigned to a topic $j$, then the number of times that word $i$ was assigned to topic $j$ becomes the weight of the directed edge from node $i$ to node $j$. These rankings were then used to weight the counts to decide which topics to keep. K-means and DBSCAN were run against the $\phi$ distributions. The number of centroids

for k-means was set to 100. For DBSCAN $\epsilon$ was set to 0.115 with the minimum number of points for a dense region as 1. We take the point closest to the centroid (Centroid), the distance to the centroid (Distance), and the distance weighted using the ranking score (Distance+Rank) to rank and choose topics to keep for k-means. For DBSCAN, we take the topic with minimal distance to all other topics in the cluster. The same experiment was performed on all nonhierarchical datasets.

### 5.5.2. Experimental Results

The algorithms were run against the $\phi$ and count matrices after 800 iterations in the topic model out of 1000 total iterations. The decision to make is to choose the best 100 out of 176 (this can be different depending on the data source and random seed) candidate topics. As shown in Figure 5 and Table 7, KnowledgeRank improves upon the existing method of counting, while k-means-based decisions and DBSCAN mostly have no effect. From an intuitive perspective, this problem is well served for KnowledgeRank. The reason why the topic model does not assign the words to the correct topic is due to another topic, not used in the generation of the corpus, that takes the assignments. By ranking the counts to topics, we can give less importance to words that belong to many different topics. These words can skew the counts and lead to incorrect topic decisions while weighting them appropriately by the amount they overlap, which ranking methods are quite good at, allowing for a better decision.

**Table 7.** Increase in topic selections for all datasets when using KnowledgeRank- and clustering-based methods over simple counting during topic modeling inference.

| | %Δ Topic Selection | | | | |
|---|---|---|---|---|---|
| | **Rank** | **Centroid** | **Distance** | **Distance+Rank** | **DBSCAN** |
| MeSH | 50 | $-70$ | $-85$ | $-66.667$ | $-87.755$ |
| CiteULike-180 | 10 | 0 | 0 | 0 | 0 |
| FAO-30 | 6.25 | 0 | 0 | 0 | 6.25 |
| SemEval-2010 | 30 | 0 | 0 | 0 | 0 |
| Reuters-21578 | 50 | 0 | 0 | 0 | 0 |

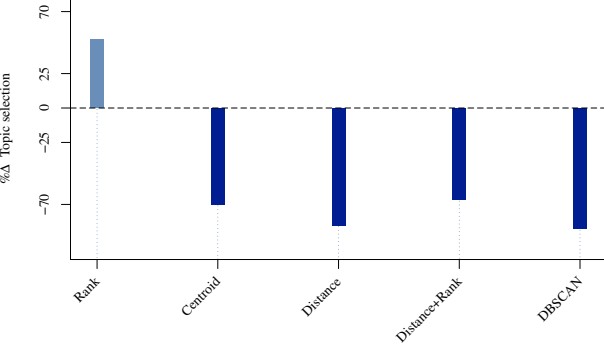

**Figure 5.** A bar chart representing increase in topic filtering decisions made during inference using KnowledgeRank- and clustering-based methods as a percentage over naive approach of simple counting of assignments to each topic using MeSH dataset.

### 5.6. Partial Knowledge

We show the utility of Rank-LDA to aid existing semisupervised topic models with partial knowledge. Rank-LDA is used to assign meaningful labels to topics that contain a large number of words that appear in the corpus but do not occur in the predetermined article-based knowledge sources. Each corpus used in evaluation consists of a subset of topics as the knowledge source and their children as the source for tokens not in the knowledge source.

5.6.1. Experimental Setup

To demonstrate how we constructed the dataset we use the MeSH corpus as an example. All other datasets were similarly constructed. From the entire MeSH hierarchy, we crawl Wikipedia for the resultant topic documents. This set consists of 20,050 found topics. We randomly select 65 topics that contain at least two direct children in the found topics from this set. We take two random direct children from the 65 parent topics for a total of 130 child topics. We then build the knowledge graph based on SNOMED CT filtered by tokens appearing in the corpus. We connect the nodes together using the graph structure of SNOMED CT and the article-based knowledge source. We add an edge between every topic node and a word existing in the corresponding article. Next, we run a modified version of the generative model for the bijective model [10] of Source-LDA with parameters of $K$, $\alpha$, $\beta$, $\mu$, $\sigma$, $D$, $D_{avg}$, as 65, $50/K$, $200/V$, 5.0, 0.0, 2000, and 500 respectively. The modification is for each word we flip an unbiased coin to decide if we are to sample from the parent topic under the Source-LDA parameters or from the raw child distributions. This process results in close to a 50/50 split between a word coming from the parent or the child. For each child word, we mark the topic assignment as that of the parent and keep the parent word as assigned to the parent topic. We then run the Rank-LDA topic model in comparison with Source-LDA, explicit Dirichlet allocation (EDA) [26], the concept topic model (CTM) [27], Sawtooth Factorial Topic Embeddings Guided Gamma Belief Network (SawETM) [4], the Variationally-Learned Recurrent Neural Topic Model (VRTM) [6], and a version of VRTM defined to utilize outside information in the form of word embeddings [55] and is evaluated as a separate model (VRTM+W2V) to determine which topic each word belongs to. Each neural topic model was implemented as described in their respective publications [7,8]. For all hierarchical datasets, we repeat the experiment as described above with the corresponding dataset topics.

5.6.2. Experimental Results

After 1000 iterations, we compare the perplexity ($\Gamma$) and classification accuracy ($\Lambda$) as a measure of goodness between the models. As is shown in Table 8, Rank-LDA outperforms all other semisupervised topic models in terms of correctly assigning each word to the correct label ($\Lambda$%), as well as perplexity. Rank-LDA similarly outperforms the baseline neural topic models as demonstrated in Table 9. The semisupervised baseline methods, Source-LDA, EDA, and CTM are limited in being restricted by their knowledge source distributions leading to a low probability of a word being assigned to the topic when it is not in the knowledge source topic. Rank-LDA rectifies this deficiency by bringing in additional outside information to connect words that may not show up in the original knowledge source article. Another interesting aspect is that Rank-LDA outperforms the neural topic models in perplexity. It is somewhat expected for Rank-LDA to outperform the baseline models in label assignment accuracy, however perplexity is a major benefit of the neural topic model over Bayesian models. We submit the reason for better performance has to do with benefit of our model coupled with the generated data. These results suggest in data that is generated under a generative model, Bayesian models can outperform neural topic models. A finding that suggests the gains in perplexity to Bayesian topic models in reported studies [7,8] may be due to the assumed generative model of the Bayesian topic models.

**Table 8.** Classification accuracy of token assignments and perplexity values for Rank-LDA, Source-LDA, EDA, and CTM using a corpus mixed evenly between parent and child topics.

|  | Rank-LDA | | Source-LDA | | EDA | | CTM | |
|---|---|---|---|---|---|---|---|---|
|  | $\Gamma$ | $\Lambda\%$ | $\Gamma$ | $\Lambda\%$ | $\Gamma$ | $\Lambda\%$ | $\Gamma$ | $\Lambda\%$ |
| MeSH | 935.8 | 62.9 | 4432.5 | 50.1 | 20,390.8 | 42 | 2040.7 | 30.7 |
| PhySH | 47.6 | 75.9 | 7262.3 | 50.2 | 14,447.9 | 41.5 | 981.2 | 39.2 |
| ACM-2012 | 3.6 | 49.9 | 7637 | 49.9 | 1481.9 | 49.9 | 200.6 | 49.9 |
| OAD-Wiki | 202.3 | 74.4 | 25,407.6 | 51.8 | 11,363.1 | 37 | 1197.9 | 12.5 |

**Table 9.** Classification accuracy of token assignments and perplexity values for Rank-LDA, SawETM, VRTM, and VRTM+W2V using a corpus mixed evenly between parent and child topics.

|  | Rank-LDA | | SawETM | | VRTM | | VRTM+W2V | |
|---|---|---|---|---|---|---|---|---|
|  | $\Gamma$ | $\Lambda\%$ | $\Gamma$ | $\Lambda\%$ | $\Gamma$ | $\Lambda\%$ | $\Gamma$ | $\Lambda\%$ |
| MeSH | 935.8 | 62.9 | 6212.5 | 1.82 | 985.1 | 1.84 | 2073 | 1.85 |
| PhySH | 47.6 | 75.9 | 2200.6 | 5.72 | 910.3 | 5.72 | 266.3 | 5.7 |
| ACM-2012 | 3.6 | 49.9 | 326.8 | 50.1 | 167.4 | 50 | 63.6 | 50 |
| OAD-Wiki | 202.3 | 74.4 | 3223.5 | 3.04 | 1225.9 | 3.04 | 1401.6 | 3.04 |

*5.7. Interpretability*

To show the how knowledge source rankings affect interpretability, we follow established crowdsourcing techniques [8] to measure interpretability of our proposed model against baseline models. The two interpretability tasks we measure are topic intrusion and word intrusion.

5.7.1. Experimental Setup

We extract the Wikipedia article for each Medline Plus article heading from the Medline Plus corpus. We add into the knowledge source, articles which are a descendant, ancestor, or no relation to a Medline Plus article heading according to MeSH. The knowledge source consists of 1000 articles and 1000 knowledge source topics. We take Medline Plus as our corpus, which consists of 961 articles. Next, we run LDA with parameters $K$, $\alpha$, $\beta$ as 100, $50/K$, $200/V$ respectively followed by Source-LDA on the corpus with $K$, $\alpha$ $\beta$, $\mu$, $\sigma$, as 100, $50/K$, $200/V$, 1.0, 0.3 respectively for 1000 iterations. Next, we run a version of Rank-LDA where we filter out 800 topics before inference (Preprocessing) and use ranking to prune topics during inference (Inference Pruning). The parameters for Rank-LDA are the same as Source-LDA with $\zeta_a$ and $\zeta_g$ both as 0.5. The graph used in Rank-LDA is built from the knowledge source and the UMLS described in the methods section. We repeat the above for ten LDA, Source-LDA, Rank-LDA, SawETM, VRTM, and VRTM+W2V runs. The neural topic models were implemented as described in Section 5.6. To generate the topic intrusion task, we choose a random run, then a random document from a set of Medline Plus article headings that do not require specialized medical knowledge. After a document is chosen, we take the two most probable topics from $\theta$ and a random selection of the least probable topics as the intruder topic. We present the user with the title of the article and the first 100 words—with the option to view the entire article. After reading the title and article, the user must identify the intrusive topic from the set of 3 topic labels. For LDA, we use the eight most common words as the topic label. In the word intrusion task, we first select a random run from LDA, Source-LDA, and Rank-LDA, then randomly choose from the following a set of easily understood and overlapping topics. From the chosen topic, we choose the four most probable words from $\phi$. The intrusive word is taken randomly from the five least probable words from $\phi$ that are also highly probable words in some other topic. The user is then presented with the topic label and asked to choose the intrusive word from the combined set of four probable and one improbable words. We filter out obscure words and topics for both the topic intrusion and word intrusion tasks.

### 5.7.2. Experimental Results

The tasks are placed on Amazon Mechanical Turk. (https://www.mturk.com/, accessed on 1 December 2021) For each task, a total of 75 questions are generated, 25 each for LDA, Source-LDA, Rank-LDA, SawETM, VRTM, and VRTM+W2V. Each task is assigned five workers. After the assignments are completed, we compare how well each model did versus the null hypothesis. For the null hypothesis, we assume a random guess. For the topic intrusive task, Rank-LDA and Source-LDA score a p-value of 0.0249 and 0.0742 with mean values of 0.448, 0.416 respectively. These scores imply significance at the 90% confidence level for both models. For the word intrusive task, we obtain p-values for both Rank-LDA and Source-LDA as less than 0.001 with mean values of 0.416 and 0.348 respectively. While there is not much interpretability gain over LDA for the topic intrusion task, there is a significant improvement in the word intrusion task (mean value of 0.272 for LDA against 0.416 for Rank-LDA). The neural topic models perform poorly on both tasks, more so than the Bayesian topic models. This findings is consistent with recent studies on neural topic models and interpretability [7]. Each task's results are plotted as a box plot in Figure 6. Each dot represents an answer whose value is set to the mean of that group. The groupings are based on the worker and topic for the word intrusion task, and worker and document for the topic intrusion task. The dashed line represents the mean of the null hypothesis.

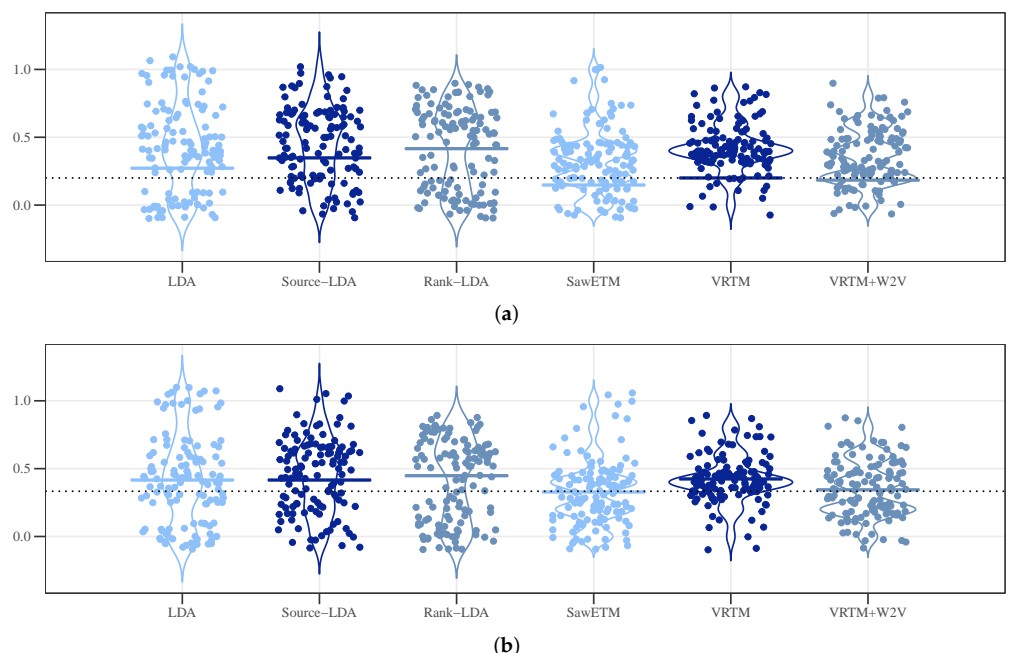

**Figure 6.** Results showing mean group answers for word intrusion task (**a**) and topic intrusion task (**b**).

## 6. Discussion

This work removes a barrier to the widespread use of semisupervised topics models. Given that we can now use any size knowledge source size as input, speed of execution is no longer an impediment. Existing semisupervised topic modelers may find our approach to be beneficial as well to reduce feasible yet slow running times back to normal speed. Additionally, we provide an alternative to counting in the inference stage, which yields better topic elimination decisions. When used as an extension in the generative model itself (Rank-LDA), we provide a topic model that improves the state-of-the art method (Source-LDA) for both perplexity and interpretability. Put together the technique presented in this paper yields improvements in three vital areas of topic modeling: (1) execution time, (2) perplexity, and (3) interpretability. We hope that by demonstrating these improvements, semisupervised topic modeling becomes more widely used.

The improvements of our approach over existing semisupervised topic models is impactful however it is far from complete. One limitation is the input size of the semisupervised input. Given the nature of PageRank, the input size is limited to the order of $10^5$. While this greatly increases the number of inputs that can be handled by our model, it is conceivable that an input size may be larger than $10^5$ and unable to be used with out model. Future work may consider a nonparametric model to handle a theoretically infinite input size. Also the ranking method we provide here may be vastly improved as input into an ensemble technique which uses the baseline methods together with other information retrieval techniques. The ensemble method approach is left as an open research area. Another limitation is the inputs used in semisupervised topic models. While generally less restrictive than supervised learning models, there are a limited amount of semisupervised data available. In our study, we utilize Wikipedia due to its completeness, but outside of Wikipedia for the general domain there are not many alternatives. This hurdle is an interesting area of future research.

Although the addition of graph-based knowledge sources was established as beneficial to various text mining tasks, it is yet to be demonstrated for inputs used in semisupervised topic modeling. Our approach represents a novel technique for representing a knowledge source article as a graph and extracting meaningful information from that graph. We also provide a technique to adding additional contextual knowledge into topic modeling. Our work is the first topic model that biases topic construction to both written word knowledge sources and graphical-based knowledge sources. It also represents the state-of-the-art technique for semisupervised topic modeling when given both graphical and text-based knowledge sources, in both perplexity and interpretability measurements; when the knowledge source input size is very large, our approach is the only feasible technique currently available.

## 7. Conclusions

This paper introduces novel methods for representing knowledge sources as graphs and ranking the nodes representing topics. These rankings can be applied to existing semisupervised topic models. When used in the preprocessing stage, KnowledgeRank is helpful to eliminate unnecessary topics. Eliminating topics before inference helps speed up the topic modeling and allows the topic model to focus on a more appropriate superset of topics. This ranking can be used during inference in place of existing elimination techniques based on counting or clustering. When used alongside semisupervised models that use an article-based knowledge source, a graph-based knowledge source improves the topic labeling. The result is better perplexity and improved interpretability.

This work is fitting to applications which depend on topic interpretability and labeling. At the University of California Los Angeles (UCLA), we are building a topic modeling visualization system for aiding primary care physicians (NIH National Library of Medicine, R21 LM011937). In this application it is paramount that topics are both interpretable and labeled, as we only give the labels across a time series as the key to understanding a patient history. The importance of basing a medical decision off a learned history, summarized by topics places a emphasis on interpretability and topic labeling over other metrics such as perplexity. This work has the potential to improve the understanding of a patient history leading to better outcomes, which could lead to drastic improvements in medical decisions and even prevent medical errors, thus saving lives. This work underscores and important application of our approach but it is not limited to this use case. We envision our approach to be beneficial for any application of topic modeling where interpretability and topic labeling are visually displayed to give end-users a quick and efficient understanding of a document.

**Author Contributions:** Conceptualization, J.W. and C.A.; methodology, J.W.; software, J.W.; validation, J.W.; formal analysis, J.W.; investigation, J.W. and C.A.; data curation, J.W; writing, J.W.; visualization, J.W.; resources, W.W.; supervision, C.A. and W.W.; funding acquisition, W.W. All authors have read and agreed to the published version of the manuscript.

**Funding:** This research received no external funding.

**Institutional Review Board Statement:** Not applicable.

**Informed Consent Statement:** Informed consent was obtained from all subjects involved in the study.

**Data Availability Statement:** Publicly available datasets were analyzed in this study. These data can be found here: https://github.com/ucla-scai/KnowledgeRank, accessed on 1 December 2021.

**Conflicts of Interest:** The authors declare no conflict of interest.

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
