# Peer review of "Knowledge Source Rankings for Semi-Supervised Topic Modeling"

_information, doi:10.3390/info13020057_

Round 1

Reviewer 1 Report

The authors raise a very interesting topic. In the first part of the article, there is a comprehensive presentation of the state of art, supported by relevant literature sources. Subsequently, the case study was presented in a very clear way, making it easier for the reader to understand the problem. The authors present in an interesting way the theoretical basis of their research and its meaning. The authors provide the obtained results together with relevant comments. Although the presented discussion and conclusions provide the added value of the results achieved by the authors, in my opinion the authors should give more praise to the importance of the practical importance of the results of their research. It is important to precisely indicate the applicability of research results not only as an academic case considered on many pages of the article, but to show: what our research gives to an ordinary user, because science is not science for itself, but is to serve people.
After these minor changes and additions, I believe that the article is worth publishing.

Reviewer 2 Report

First of all, congratulations on submitting the manuscript to the Information journal. Here are my suggestions on how the manuscript could be improved:

1) At the end of the manuscript abstract, some numerical values could be given of how much better the authors proposed approach compared to the analyzed.

2) Sometimes, the authors use the "our method", sometimes the "approach". I would say that it is a little bit confusing, so it would be better to use just one definition.

3) At least I do not like such an early beginning of the Introduction. Because it starts more look like the Related works, when starts analyzing methods, effectiveness, etc. I would suggest writing at least a small paragraph or a few sentences about text analysis, its importance nowadays, and so on. It would be good to write about the areas where it could be applied, what tasks help to solve, etc. There are different approaches where LDA and LSA are combined with other methods in text analysis, so it could also be mentioned how it can help in various text analyses: https://doi.org/10.15388/22-INFOR473. Now is given a very narrow view or area, the introduction usually shows a bigger scope, to show that authors know what is happening around in this area.

4) 30 line, Figure 2 is so far away, and why the first reference is given to Figure 2, but not the 1?

5) 42 line, as well as in all text, I would suggest using the math index by d1 (1 as an index of d). Later in the text, it even changes to the index, so needs to use the same style.

6) 44 line, sometimes used "-", sometimes "[1–3]" or "1-4", maybe also would be good to use the same signs.

7) 172 line, something wrong with the index 6,7, becomes a 67

8) I suggest that separating Introduction and Related works much better because now the introduction is so long, a little bit confusing to read it looks more like methods description, but not the introduction. I recommend rewriting it, making some additional sections, and more structure in the manuscript needs.

9) At the end of the Introduction, could be written the structure of manuscript.  What is presented in Sections 2, 3, 4, etc.

10) 247-253, in the same paragraph a few times the same reference, I don't think it needs to be always written again, and again.

11) Usually in all MDPI journals, the table names are over the table.

12) Table 4,5, it would be better to write full names instead of KG, K, etc. because we need again to look in the text to remember what it means, so the table is not so clear.

13) It is not clear or maybe just I do not understand, so if we have a text, does any preprocessing of the text have been done? I mean suppose that we have a set of texts, so there is a lot of unnecessary information like numbers, etc., which also will be considered as a topic in some methods. Does it was filtered, or do the methods already have inside of them some algorithms which ignore it.

14) The formatting of the references (not all) does not correspond to the format of MDPI journals.

The manuscript is an interesting but very confusing text. There are many formulas, some of which look unimportant. Also, there are many small errors left, like different marks, notation, the definition of the same word, so the general text of all manuscripts must be improved. Conclusions are very weak, need to give some numbers, and concludes better the all researched results performed.

Good luck, I hope the authors improve the text and that the manuscript will be submitted to the journal successfully.

Round 2

Reviewer 2 Report

Thank you for taking suggestions into account. Good luck with a final submission.